# Predictability of HOMA-IR for Gestational Diabetes Mellitus in Early Pregnancy Based on Different First Trimester BMI Values

**DOI:** 10.3390/jpm13010060

**Published:** 2022-12-28

**Authors:** Yanbei Duo, Shuoning Song, Yuemei Zhang, Xiaolin Qiao, Jiyu Xu, Jing Zhang, Zhenyao Peng, Yan Chen, Xiaorui Nie, Qiujin Sun, Xianchun Yang, Ailing Wang, Wei Sun, Yong Fu, Yingyue Dong, Zechun Lu, Tao Yuan, Weigang Zhao

**Affiliations:** 1Department of Endocrinology, Key Laboratory of Endocrinology of Ministry of Health, Peking Union Medical College Hospital, Chinese Academy of Medical Science and Peking Union Medical College, Beijing 100730, China; 2Department of Obstetrics, Haidian District Maternal and Child Health Care Hospital, Beijing 100080, China; 3Department of Obstetrics, Beijing Chaoyang District Maternal and Child Health Care Hospital, Beijing 100026, China; 4Core Facility of Instrument, Institute of Basic Medical Sciences, Chinese Academy of Medical Sciences, School of Basic Medicine, Peking Union Medical College, Beijing 100005, China; 5Department of Laboratory, Haidian District Maternal and Child Health Care Hospital, Beijing 100080, China; 6Department of Dean’s Office, Haidian District Maternal and Child Health Care Hospital, Beijing 100080, China; 7Department of Clinical Laboratory, Beijing Chaoyang District Maternal and Child Health Care Hospital, Beijing 100026, China; 8National Center for Women and Children’s Health, China CDC, Beijing 100013, China

**Keywords:** HOMA-IR, gestational diabetes, body mass index, early pregnancy, predictability

## Abstract

**Objective**: To investigate the ability of homeostasis model assessment of insulin resistance (HOMA-IR) in early pregnancy for predicting gestational diabetes mellitus (GDM) in Chinese women with different first-trimester body mass index (FT-BMI) values. **Methods**: Baseline characteristics and laboratory tests were collected at the first prenatal visit (6–12 weeks of gestation). GDM was diagnosed by a 75 g oral glucose tolerance test (OGTT) at 24–28 weeks of gestation. Partial correlation analysis and binary logistic regression were applied to identify the association between HOMA-IR and GDM. The cutoff points for predicting GDM were estimated using receiver operating characteristic (ROC) curve analysis. **Results**: Of the total of 1343 women, 300 (22.34%) were diagnosed with GDM in the 24–28 weeks of gestation. Partial correlation analysis and binary logistic regression verified HOMA-IR as a significant risk factor for GDM in the normal weight subgroup (FT-BMI < 24 kg/m^2^) (adjusted OR 2.941 [95% CI 2.153, 4.016], P < 0.001), overweight subgroup (24.0 kg/m^2^ ≤ FT-BMI < 28.0 kg/m^2^) (adjusted OR 3.188 [95% CI 2.011, 5.055], P < 0.001), and obese subgroup (FT-BMI ≥ 28.0 kg/m^2^) (adjusted OR 9.415 [95% CI 1.712, 51.770], *p* = 0.01). The cutoff values of HOMA-IR were 1.52 (area under the curve (AUC) 0.733, 95% CI 0.701–0.765, *p* < 0.001) for all participants, 1.43 (AUC 0.691, 95% CI 0.651–0.730, *p* < 0.001) for normal weight women, 2.27 (AUC 0.760, 95% CI 0.703–0.818, *p* < 0.001) for overweight women, and 2.31 (AUC 0.801, 95% CI 0.696–0.907, *p* < 0.001) for obese women. **Conclusions**: Increased HOMA-IR in early pregnancy is a risk factor for GDM, and HOMA-IR can be affected by body weight. The cutoff value of HOMA-IR to predict GDM should be distinguished by different FT-BMI values.

## 1. Introduction

Gestational diabetes mellitus (GDM) is a condition of glucose intolerance that is first recognized during pregnancy and is diagnosed during the second trimester [1]. With the changes in lifestyle and nutrition, the prevalence of GDM has been increasing in recent years. According to the International Association of Diabetes and Pregnancy Study Group (IADPSG) criteria, the incidence of GDM has reached 18.3–24.24% in China [2,3,4]. GDM is associated with an increased risk of several perinatal complications, and it may influence the long-term health of both mothers and their offspring [5]. High pre-pregnancy body mass index (preBMI) and excessive weight gain during pregnancy are risk factors for the development of GDM [6]. However, women with normal preBMI and appropriate gestational weight gain may also develop GDM, indicating that other factors are involved in the pathogenesis of GDM.

Homeostasis model assessment of insulin resistance (HOMA-IR) is an indicator of insulin resistance in diabetic patients [7]. With the change in maternal hormone levels in the first trimester of pregnancy, slight insulin resistance and increased insulin secretion may affect fetal growth and energy reserves [8]. GDM often occurs when pancreatic beta cells are unable to produce sufficient insulin to adapt to physiological changes. Several risk factors play a role in the development of insulin resistance and GDM, including age, race, obesity, and parity [6,9,10,11]. Although HOMA-IR is well correlated with insulin resistance, there is no standardized cutoff value of HOMA-IR for the identification of insulin resistance. It has been reported that 5 kg of weight gain results in a 1.6-fold increase in HOMA-IR, indicating that BMI is closely related to HOMA-IR [12]. Therefore, HOMA-IR thresholds should be adjusted depending on different preBMI or first-trimester BMI (FT-BMI) values to predict GDM more accurately. 

Because of the complications caused by GDM and the health concerns about both mothers and offspring, the diagnosis of GDM should be confirmed as early as possible. So far, there are still differences in the diagnosis of GDM. A one-step strategy of 75 g oral glucose tolerance test (OGTT) is adopted in our country, while in other countries (especially in European countries), the diagnosis of GDM is confirmed by the two-step strategy of 100 g OGTT [1]. There are neither uniform criteria for the diagnosis of GDM by HOMA-IR in early pregnancy, nor is there a threshold of HOMA-IR based on different BMI. In addition, the cutoff value of HOMA-IR for Chinese women in early pregnancy to predict GDM has not been confirmed. Based on the above, we evaluated HOMA-IR in the first trimester of gestation among Chinese women with different FT-BMI values to determine the optimal cutoff value of HOMA-IR in early pregnancy for predicting GDM. 

## 2. Materials and Methods

This study was part of an ongoing prospective double-center observational cohort study started in 2019, which was conducted at Haidian District Maternal and Child Health Care Hospital and Chaoyang District Maternal and Child Health Care Hospital (Beijing, China) (clinical trial number: NCT03246295). 

The Ethics Committees of both participating centers approved the study protocol. Written informed consent was obtained from each participant, and the study was performed in accordance with the Declaration of Helsinki as revised in 2013.

### 2.1. Study Participants

Baseline characteristics of all participants were collected at the first prenatal visit in the first trimester of pregnancy (between 6 and 12 weeks). The inclusion criteria were as follows: (1) gestation less than 12 weeks at the first visit; (2) without known impaired glucose tolerance or diabetes before pregnancy; (3) no medication used, except for vitamins; (4) acceptance of participation in the study and signature on the consent form; and (5) participation in regular follow-ups. The exclusion criteria were as follows: (1) non-singleton pregnancy; (2) severe systemic diseases (e.g., liver disease, kidney failure, cardiac insufficiency, autoimmune disease, or hematological disease); and (3) inability to understand and complete the study. Because all participants were enrolled during early pregnancy, professional advice was given by clinicians from the first prenatal visit to delivery. 

### 2.2. Clinical and Laboratory Measurements

Detailed clinical and demographic data, including age, FT-BMI, gravidity, parity, pregnancy history, and family history of diabetes, were collected at the first visit (6–12 weeks of gestation). The height, body weight, systolic blood pressure (SBP), and diastolic blood pressure (DBP) were measured at each follow-up visit. The BP was measured twice at 5 min intervals using an automatic BP monitor. BMI was calculated using the following formula: BMI = weight/height^2^. According to China’s standards on weight management for overweight or obese individuals, BMI ≥ 24 kg/m^2^ was defined as overweight, while BMI ≥ 28 kg/m^2^ was defined as obese [13]. 

At the first prenatal visit within 6–12 weeks of pregnancy, fasting blood samples were collected to measure the fasting blood glucose (FBG) level, fasting insulin (FI) level, fasting C-peptide (FCP) level, lipid profile [total cholesterol (TC), triglyceride (TG), high-density lipoprotein cholesterol (HDL-C), low-density lipoprotein cholesterol (LDL-C)], hepatic function and renal function. All blood samples were collected in the morning after a 10 to 12 h overnight fast followed by the doctor’s advice. HOMA-IR, which indicates insulin resistance, was calculated using the following formula: HOMA-IR = (FBG [mmol/L]) × FI [µU/mL]/22.5). Homeostasis model assessment of beta cell function (HOMA-β), which indicates beta cell function, was calculated using the following formula: HOMA-β= (20 × FI [µU/mL]/(FBG [mmol/L]–3.5)) [14]. Insulin sensitivity, as indicated by the quantitative insulin sensitivity check index from insulin (QUICKI), was calculated using the following formula: QUICKI = 1/(log_10_ FPG [mg/dL] + log_10_ FI [µU/mL]). HOMA-β was lost in 64 participants because the FBG levels were 3.5 mmol/L or less.

All participants were screened for GDM by a 75 g oral glucose tolerance test (OGTT) during weeks 24 and 28 of gestation. GDM was diagnosed according to the IADPSG criterion in 2010 [15]. Overall, 1343 pregnant women with baseline characteristics (6–12 weeks of pregnancy) and 75 g OGTT at 24–28 weeks of gestation were enrolled in the present study. All available clinical and laboratory data were recorded and verified by two researchers at the same time. 

The following pregnancy outcome data were collected: gestational age at birth, type of delivery, birth weight, and 10 min Apgar score. Macrosomia was defined as birth weight >4 kg, and low birth weight was defined as birth weight <2.5 kg. Preterm delivery was defined as delivery before gestational week 37 [16,17]. Due to the coronavirus pandemic since 2019, 229 of the 1343 participants (17.05%) dropped out or were lost to follow-up after 28 weeks of gestation. 

### 2.3. Statistical Analysis

Clinical characteristics and pregnancy outcomes were compared between the GDM group and normal glucose tolerance (NGT) group. In order to compare the incidence of GDM and insulin sensitivity in pregnant women with different FT-BMI values, the participants were divided into the following three groups: normal weight (BMI < 24 kg/m^2^), overweight (24 kg/m^2^ ≤ BMI < 28 kg/m^2^), and obesity (BMI ≥ 28 kg/m^2^). Continuous variables are presented as the mean ± SD if normally distributed and as medians (interquartile range) otherwise, and categorical variables are presented as percentages. The χ^2^ test (or Fisher’s exact test in case of small frequencies) was used for comparing categorical variables between groups. As appropriate, two-sample Student *t* tests and a Mann–Whitney U test and Kruskal–Wallis test was applied for group comparisons of continuous variables. *p* < 0.05 (two-tailed) was considered significant. Partial correlation analysis was applied to identify correlations between HOMA-IR, HOMA-β, and QUICKI during 6–12 weeks of gestation and 75 g OGTT during 24–28 weeks of pregnancy. Binary logistic regression analysis was used to evaluate the association between HOMA-IR and GDM. Receiver operating characteristic (ROC) curve analysis was used to determine the optimal cutoff value of HOMA-IR for the prediction of GDM. Statistical significance was inferred from two-sided *p* values <0.05. Statistical analyses were performed using the SPSS statistical program (version 25.0; IBM Corporation, Armonk, NY, USA). 

## 3. Results

In the present study, the incidence of GDM was 22.34% (300/1343). Table 1 shows the clinical and laboratory characteristics of pregnant women with and without GDM. Compared to women with NGT, women with GDM were relatively older and with higher FT-BMI. Family history of diabetes and adverse pregnancy history were both higher in the GDM group with statistical significance (*p* < 0.05). Almost all laboratory data in early pregnancy were statistically different (*p* < 0.05) between the two groups except HOMA-β; compared to the NGT group, HOMA-IR was significantly higher in the GDM group, while QUICKI was significantly lower in the GDM group (*p* < 0.001). Blood pressure in the third trimester, preterm delivery rate, and newborn birth weight did not differ significantly between the two groups. 

In order to explore the difference among women with different FT-BMI values, the participants were stratified into the following three subgroups: normal weight (FT-BMI < 24 kg/m^2^, 1066 [79.37%]), overweight (24 kg/m^2^ ≤ FT-BMI < 28 kg/m^2^, 210 [15.64%]) and obesity (FT-BMI ≥ 28 kg/m^2^, 67 [4.99%]) (Appendix A). The incidence of GDM increased with FT-BMI, which was 17.92% in the normal weight subgroup, 34.76% in the overweight subgroup, and 53.73% in the obesity subgroup, separately (*p* < 0.01). The glycometabolic data in early pregnancy differed significantly among the three subgroups with higher FCP, FI, HOMA-IR, and HOMA-β but lower QUICKI in women with higher FT-BMI (*p* < 0.001). Compared with normal-weight women, the cholesterol levels (TC, TG, HDL-C, LDL-C) differed significantly in overweight and obese women. The gestational weight gain (GWG) was the highest (13.0 kg [11.0–16.0]) in the normal weight subgroup but the lowest (8.5 kg [4.6–12.0]) in the obesity subgroup, which indicated that weight gain was greater in the normal weight women and less in the obese women. 

For the purpose of exploring the relationship between FT-BMI and glycometabolism, the pregnant women in this study were divided into the following four subgroups: normal weight (FT-BMI < 24.0 kg/m^2^) with NGT (n = 875); normal weight (FT-BMI < 24.0 kg/m^2^) with GDM (n = 191); overweight (FT-BMI ≥ 24.0 kg/m^2^) with NGT (n = 168); and overweight (FT-BMI ≥ 24.0 kg/m^2^) with GDM (n = 109) (Appendix A). FBG, FCP, FI, HOMA-IR, and lipid profile (TC, TG, and LDL-C) were the lowest in participants who were categorized as the normal weight with NGT and the highest in overweight women with GDM (*p* < 0.001). Among pregnant women with NGT, FBG, FCP, FI, HOMA-IR, and HOMA-β were significantly higher, but QUICKI was significantly lower in overweight women compared with normal weight (*p* < 0.01). Similar results were found in pregnant women with GDM. 

Partial correlation analysis was used to detect the correlation between insulin resistance in early pregnancy and 75 g-OGTT at 24–28 weeks of gestation. After adjusting for age, FT-BMI, GWG, and serum lipid profile, HOMA-IR, and QUICKI in early pregnancy were significantly correlated with fasting, 1 h 75 g OGTT and 2 h 75 g OGTT (*p* < 0.05). HOMA-IR showed the highest correlation with fasting 75 g OGTT (r = 0.270, *p* < 0.01) (Table 2). 

Binary logistic regression was applied to explore the association between HOMA-IR in early pregnancy and GDM (Table 3). After adjusting for age, FT-BMI, adverse pregnancy history, and family history of diabetes, GWG, and serum lipid profile in early pregnancy, HOMA-IR remained independently associated with GDM (adjusted OR 2.966 [95% CI 2.306–3.814], *p* < 0.001). Because HOMA-IR can be influenced by body weight, the participants were divided into the following subgroups according to FT-BMI: FT-BMI < 24.0 kg/m^2^, 24.0 kg/m^2^ ≤FT-BMI < 28.0 kg/m^2^, and FT-BMI ≥ 28.0 kg/m^2^. After adjusting for the above confounding factors, HOMA-IR was still a risk factor for GDM (for FT-BMI < 24 kg/m^2^ subgroup, adjusted OR 2.941, 95% CI 2.153–4.016, *p* < 0.001; for 24.0 kg/m^2^ ≤FT-BMI < 28.0 kg/m^2^ subgroup, adjusted OR 3.188, 95% CI 2.011–5.055, *p* < 0.001; for FT-BMI ≥ 28.0 kg/m^2^ subgroup, adjusted OR 9.415, 95% CI 1.712–51.770; *p* = 0.01). 

ROC curves were used to evaluate the predictive value of HOMA-IR for GDM (Figure 1). For all participants in this study, the cutoff value of HOMA-IR to predict GDM was 1.52 (sensitivity of 63% and specificity of 71%) with an AUC value of 0.733 (95% CI 0.701–0.765; *p* < 0.001) (Figure 1A). Different cutoff values were determined according to FT-BMI. For participants with FT-BMI < 24 kg/m^2^, the cutoff value was 1.43 (sensitivity of 56% and specificity of 71%) (Figure 1B) with an AUC value of 0.691 (95% CI 0.651–0.730; *p* < 0.001). For participants with 24.0 kg/m^2^ ≤FT-BMI < 28.0 kg/m^2^, the cutoff value was 2.27 (sensitivity of 56% and specificity of 83%) with an AUC value of 0.760 (95% CI 0.703–0.818; *p* < 0.001) (Figure 1C); For participants with FT-BMI ≥ 28.0 kg/m^2^, the cutoff value was 2.31 (sensitivity of 75% and specificity of 76%) with an AUC value of 0.801 (95% CI 0.696–0.907; *p* < 0.001) (Figure 1D). 

## 4. Discussion

In this prospective cohort study, pregnant women were divided into different groups based on FT-BMI to identify the optimal cutoff values of HOMA-IR in early pregnancy for predicting GDM. To the best of our knowledge, this is the first study to explore the ability of HOMA-IR to predict GDM based on different FT-BMI values. 

Insulin resistance is defined as reduced responsiveness to high physiological insulin levels in insulin-targeting tissues, and it is considered an important pathogenesis of T2DM [18]. Similar pathophysiological mechanisms exist between GDM and T2DM. Beta cell dysfunction with chronic insulin resistance during pregnancy results in beta cell impairment and tissue insulin resistance, which represent critical components of the pathophysiology of GDM [8]. Although the hyperinsulinemic-euglycemic clamp is the gold-standard method to assess insulin resistance, it is rarely used in the clinic due to its complexity and cost [19]. Several surrogate estimates have been proposed for assessing insulin sensitivity, including HOMA-IR, HOMA-β, and QUICKI [20,21]. However, the threshold of HOMA-IR for evaluating insulin resistance has not been unified because of various associated factors of HOMA-IR, including age, sex, race, and weight [22,23,24]. HOMA-IR is strongly correlated with body weight as there is a significant difference in HOMA-IR between normal weight and obese individuals [25]. Similarly, the present study demonstrated that HOMA-IR increased with increasing FT-BMI (17.92% in normal-weight women 17.92%, 34.76% in overweight women, and 53.73% in obese women), and HOMA-IR was even higher in overweight women with NGT compared with normal weight women with GDM (Appendix A). Therefore, these findings suggested that HOMA-IR is highly correlated with body weight, so the cutoff value of HOMA-IR for identifying GDM should be distinguished according to different FT-BMI values. 

Many previous studies discussed the threshold of HOMA-IR in the diagnosis of GDM. Paracha et al. reported that HOMA-IR > 2 could replace 75 g OGTT as a screening tool for GDM at 24–28 weeks of gestation [26]. However, due to the perinatal complications caused by hyperglycemia, GDM should be identified as early as possible. Alptekin et al. and Ozcimen et al. reported the cutoff value of HOMA-IR for diagnosing GDM in the first trimester, but they did not divide pregnant women by different BMI values [27,28]. In addition, Song et al. provided a novel surrogate index of HOMA-IR. However, the formula included pre-pregnancy BMI, FBG, and lipid profiles, which were relatively complicated and with poor practicability [29]. In the present prospective cohort study, the cutoff value of HOMA-IR for predicting GDM in early pregnancy was determined based on FT-BMI categories, which provided an individualized assessment of HOMA-IR for pregnant women. Although body weight gain during pregnancy was a significant risk factor for GDM, all pregnant women in this cohort received standard pregnancy guidance with GWG within reasonable limits. HOMA-IR was increased with FT-BMI, and ROC curve analysis in this study determined different thresholds of HOMA-IR for predicting GDM according to different FT-BMI, with relatively high AUC values. 

In the present study, the cutoff values of HOMA-IR in the first trimester for predicting GDM were 1.52, 1.43, 2.27, and 2.31 for all participants, normal weight women, overweight women, and obese women, respectively. A previous study has identified HOMA-IR by different body weights and found that the 75th percentile of HOMA-IR is 1.68 in normal-weight individuals and 3.42 in obese individuals [25]. However, HOMA-IR may be affected by race, and the participants of these previous studies were all Caucasians [23]. Several published studies have investigated HOMA-IR in Asian populations and determined the average HOMA-IR in healthy males and females [22,30,31]. To date, there is no study focused on the relationship between HOMA-IR and body weight in pregnant women. Thus, the present study provides a more precise cutoff value of HOMA-IR for predicting GDM in pregnant Chinese women with different FT-BMI values. 

Similar to other reports on the Chinese population, the overall prevalence of GDM in the present study was 22.34% [3,4]. However, the incidence of GDM varied among different FT-BMI values, with 17.92% in the normal weight group, 34.76% in the overweight group, and 53.73% in the obese group (Appendix A). FT-BMI is considered an independent risk factor for GDM, and previous studies have reported that overweight women are almost twice as likely to develop GDM as normal-weight women [32]. Similarly, another study has reported that the incidence of GDM is 18% in non-obese women and 32% in obese women [33]. Casagrande et al. reported that the prevalence of GDM is 4.7%, 6.5%, and 10.5% in normal weight, overweight and obese groups, respectively, indicating that the increasing incidence of GDM is associated with higher body weight [34]. These above-mentioned studies defined overweight as preBMI ≥ 25 kg/m^2^ and obesity as preBMI ≥ 30 kg/m^2^. In the present study, however, the criteria of overweight and obesity were more suitable for the Chinese population (overweight defined as BMI ≥ 24 kg/m^2^ and obesity defined as BMI ≥ 28 kg/m^2^) [13]. Due to the incidence of GDM increasing with FT-BMI, overweight and obese women should be given appropriate guidance during pregnancy to prevent GDM and other perinatal complications. 

In the present study, a proportion of women were diagnosed with GDM with normal FT-BMI, while a percentage of overweight women did not develop GDM, which suggested that other mechanisms may be involved in the pathogenesis of GDM. Some studies have examined the relevance between variants of genes and the risk of GDM. Zhang et al. performed a systematic review and found that variants in seven genes are significantly associated with GDM, including *TCF7L2*, *GCK*, *KCNJ11*, *CDKAL1*, *IGF2BP2*, *MTNR1B,* and *IRS1*, with the first six genes related to insulin secretion and *IRS1* related to insulin resistance [35]. Interestingly, all seven genes have been previously identified to be related to T2DM risk, which suggests a relationship between GDM and T2DM [36]. Some studies have discovered that other genes also contribute to the onset of GDM, such as *HNF4A* and *PPARG* [37,38]. Thus, these findings present a novel perspective on the underlying pathophysiology of GDM. 

The present study focused on insulin sensitivity and resistance in the first trimester of gestation. HOMA-IR and the quantitative insulin sensitivity check have been recommended in many institutions, but there is no commonly accepted HOMA-IR cutoff value. The standardized insulin assay was not available in the past few years; however, the technique of serum insulin test has become more and more popular for clinical use in recent years, and the diagnosis of GDM is easier and earlier by HOMA-IR in early pregnancy compared with 75 g OGTT during the second trimester. Because HOMA-IR can be affected by body weight, it is necessary to characterize pregnant women according to different BMI, which provides individualized thresholds for pregnant women. Once the HOMA-IR of pregnant women reaches the above criteria, the diagnosis of GDM can be confirmed, and measurements should be taken to reduce perinatal complications. 

Several limitations inevitably exist in the present study. Firstly, a percentage of pregnant women failed to be followed up to delivery (236 of 1388, 17.0%), perhaps because of the coronavirus pandemic restricting regular follow-ups. However, all participants in our study were not infected with COVID-19 under the dynamic zero-COVID policy and strategies executed by our country, and all participants were followed up to 24–28 weeks of gestation, which did not affect the results. Secondly, the sample size of the obesity subgroup (preBMI ≥ 28 kg/m^2^) was relatively small (67/1343, 4.99%), but the ROC-AUC value was the best among the four subgroups. Further studies may devote to exploring the threshold of HOMA-IR in early pregnancy in obese women. Thirdly, no data were available for glucose metabolism in newborns and postpartum pregnant women. 

In conclusion, the present study demonstrated that increased HOMA-IR in early pregnancy is a risk factor for GDM and is significantly influenced by body weight. Dividing HOMA-IR by BMI categories increases the accuracy of the predictive value of HOMA-IR for GDM. In the present study, the cutoff values of HOMA-IR in early pregnancy to predict GDM was 1.52 for all participants, 1.43 for normal-weight women (FT-BMI < 24 kg/m^2^), 2.27 for overweight women (24.0 kg/m^2^ ≤FT-BMI < 28.0 kg/m^2^), and 2.31 for obese women (FT-BMI ≥ 28.0 kg/m^2^). Because GDM is currently diagnosed at gestational weeks 24–28, these findings may help clinicians to identify GDM in the first trimester. If the HOMA-IR of a pregnant woman was higher than the cutoff point in early pregnancy, appropriate strategies should be taken to reduce GDM and other perinatal complications. Furthermore, a proportion of pregnant women with normal body weight also developed GDM, suggesting that other pathophysiological mechanisms exist in GDM. 

## Figures and Tables

**Figure 1 jpm-13-00060-f001:**
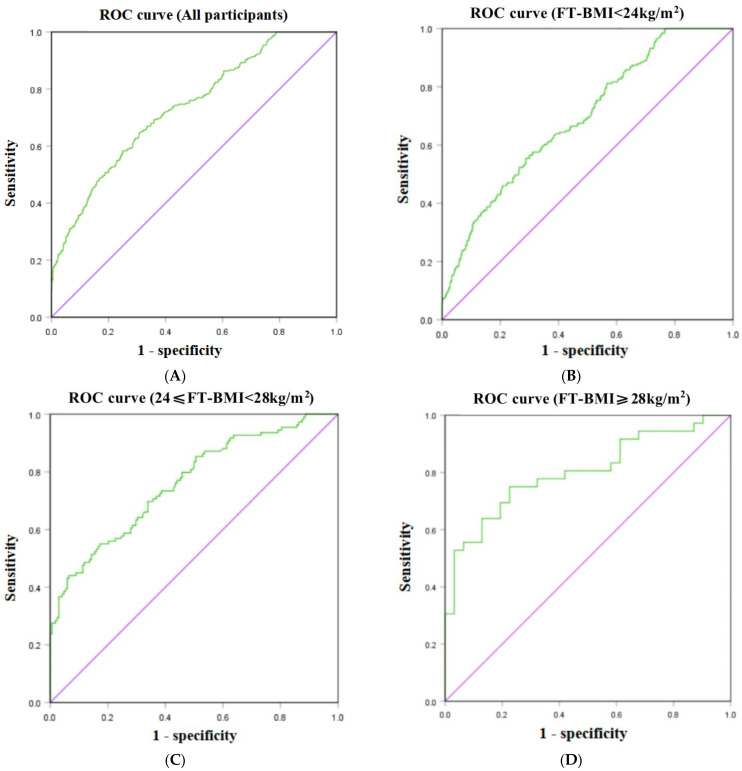
Ability of HOMA-IR in early pregnancy for predicting GDM in all participants (**A**), FT-BMI < 24 kg/m^2^ subgroup (**B**), 24 ≤ FT-BMI < 28 kg/m^2^ subgroup (**C**), and FT-BMI ≥ 28.0 kg/m^2^ subgroup (**D**).

**Table 1 jpm-13-00060-t001:** Baseline characteristics and pregnancy outcomes of pregnant women.

Characteristics		NGT (n = 1043)	GDM (n = 300)	*p*-Value
Maternal baseline information				
Age (years)		30 (28, 33)	32 (29, 35)	<0.001 **
FT-BMI (kg/m^2^)		21.34 (19.72, 23.23)	22.80 (20.57, 25.09)	<0.001 **
Gravidity	1	564 (54.07%)	156 (52.00%)	0.555
	2	283 (27.13%)	84 (28.00%)	
	≥3	196 (18.79%)	60 (20.00%)	
Parity	0	743 (71.24%)	202 (67.33%)	0.257
	1	284 (27.23%)	90 (30.00%)	
	≥2	16 (1.53%)	8 (2.67%)	
Family history of diabetes		140 (13.42%)	55 (18.33%)	0.033 *
History of adverse pregnancy ^a^		176 (16.87%)	66 (22.0%)	0.042 *
GWG (kg)		13.0 (11.0, 16.0)	11.75 (9.0, 14.0)	<0.001 **
Laboratory data between 6–12 weeks of pregnancy				
FBG (mmol/L)		4.5 (4.2, 4.9)	4.8 (4.4, 5.2)	<0.001 **
FCP (ng/ml)		0.79 (0.60, 1.04)	1.02 (0.77, 1.33)	<0.001 **
FI (uU/ml)		6.0 (4.2, 8.2)	8.3 (5.7, 11.9)	<0.001 **
HOMA-IR		1.20 (0.85, 1.66)	1.76 (1.15, 2.43)	<0.001 **
HOMA-β		123.64 (75.79, 202.00)	128.89 (85.45, 225.45)	0.103
QUICKI		0.37 (0.35, 0.39)	0.35 (0.33, 0.37)	<0.001 **
TC (mmol/L)		3.95 (3.49, 4.48)	4.20 (3.64, 4.74)	0.006 **
TG (mmol/L)		0.83 (0.64, 1.08)	1.00 (0.76, 1.35)	<0.001 **
HDL-C (mmol/L)		1.48 (1.28, 1.68)	1.45 (1.23, 1.61)	0.007 **
LDL-C (mmol/L)		1.98 (1.66, 2.40)	2.22 (1.77, 2.73)	<0.001 **
75 g OGTT between 24–28 weeks of pregnancy				
Fasting glucose OGTT (mmol/L)		4.60 (4.39, 4.78)	5.15 (4.83, 5.31)	<0.001 **
1 h glucose OGTT (mmol/L)		7.24 (6.28, 8.16)	9.71 (8.30, 10.65)	<0.001 **
2 h glucose OGTT (mmol/L)		6.24 (5.58, 6.91)	7.98 (6.95, 8.97)	<0.001 **
Characteristics in the third trimester of pregnancy				
SBP (mmHg)		118.0 (110.0, 120.0)	118.0 (110.0, 120.0)	0.923
DBP (mmHg)		72.0 (70.0, 80.0)	72.0 (70.0, 80.0)	0.916
Preterm delivery ^b^		29 (3.35%)	15 (6.02%)	0.057
Newborn birth weight (kg)	<2500 g	28 (3.24%)	8 (3.21%)	0.985
	>2500 g, <4000 g	812 (93.87%)	230 (92.37%)	
	≥4000 g	25 (2.89%)	11 (4.42%)	

Abbreviations: NGT, normal glucose tolerance; GDM, gestational diabetes mellitus; FT-BMI, first trimester body mass index; GWG, gestational weight gain; FBG, fasting blood glucose; FCP, fasting C-peptide; FI, fasting insulin; HOMA-IR, homeostasis model assessment of insulin resistance; HOMA-β, homeostasis model assessment of β-cell function; QUICKI, quantitative insulin sensitivity check index from insulin; TC, total cholesterol; TG, triglyceride; HDL-C, high-density lipoprotein cholesterol; LDL-C, low-density lipoprotein cholesterol; SBP, systolic blood pressure; DBP, diastolic blood pressure. ^a^: Defined as embryo damage, spontaneous abortion or preterm delivery in previous pregnancy. ^b^: Defined as delivery less than 37 completed weeks. Data are presented as n (%) or median (interquartile range). * *p* < 0.05 and ** *p* < 0.01.

**Table 2 jpm-13-00060-t002:** Partial correlation analysis of insulin sensitivity in early pregnancy and serum glucose at gestational weeks 24–28 ^a^.

	HOMA-IR	HOMA-β	QUICKI	Fasting Glucose OGTT	1 h Glucose OGTT	2 h Glucose OGTT
HOMA-IR	1					
HOMA-β	0.125 **	1				
QUICKI	−0.860 **	−0.131 **	1			
Fasting glucose OGTT	0.270 **	−0.093 **	−0.196 **	1		
1 h glucose OGTT	0.202 **	−0.033	−0.128 **	0.318 **	1	
2 h glucose OGTT	0.169 **	−0.038	−0.112 **	0.286 **	0.604 **	1

^a^: adjusted for age, FT-BMI, gestational weight gain and serum lipid profile (total cholesterol, triglyceride, high-density lipoprotein cholesterol and low-density lipoprotein cholesterol). ** *p* ˂ 0.01.

**Table 3 jpm-13-00060-t003:** Association of HOMA-IR in early pregnancy with GDM.

GDM	HOMA-IR of All Participants	HOMA-IR (FT-BMI < 24 kg/m^2^)	HOMA-IR (24.0 kg/m^2^ ≤FT-BMI < 28.0 kg/m^2^)	HOMA-IR (FT-BMI ≥ 28.0 kg/m^2^)
OR (95% CI)	3.282 (2.708, 3.976)	3.032 (2.354, 3.904)	3.155 (2.247, 4.430)	3.863 (1.801, 8.286)
*p* value	˂0.001 **	˂0.001 **	˂0.001 **	0.001 **
aOR1 (95% CI)	2.980 (2.423, 3.666) ^a^	2.927 (2.247, 3.812) ^a^	3.214 (2.244, 4.604) ^a^	4.649 (1.785, 12.109) ^a^
*p* value	˂0.001 **	˂0.001 **	˂0.001 **	0.002 **
aOR2 (95% CI)	2.966 (2.306, 3.814) ^b^	2.941 (2.153, 4.016) ^b^	3.188 (2.011, 5.055) ^b^	9.415 (1.712, 51.770) ^b^
*p* value	˂0.001 **	˂0.001 **	˂0.001 **	0.01 *

^a^: adjusted for age, FT-BMI, family history of diabetes and history of adverse pregnancy (defined as embryo damage, spontaneous abortion or preterm delivery). ^b^: a+ adjusted for gestational weight gain (GWG) and lipid profile (total cholesterol, triglyceride, high-density lipoprotein cholesterol and low-density lipoprotein cholesterol). * *p* ˂ 0.05 and ** *p* ˂ 0.01.

## Data Availability

Not applicable.

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
