# Peer review of "Predictability of HOMA-IR for Gestational Diabetes Mellitus in Early Pregnancy Based on Different First Trimester BMI Values"

_jpm, 2022, doi:10.3390/jpm13010060_

Round 1

Reviewer 1 Report

This is a comprehensive and overall clearly presented study showing that HOMA-IR is increased in early pregnancy in patients who subsequently develop GDM. and could be used in early pregnancy to predict this. The authors specifically address the impact of BMI in early pregnancy on HOMA IR and implications for predicting GDM.

There are some issues to address to improve this paper.

The term pre-pregnancy BMI is used but the BMI is estimated in early pregnancy in the first trimester, not before pregnancy, so the term preBMI is not strictly correct. First trimester BMI is a better term and this should be used instead of preBMI.

Abstract - Results suggest rephrase this to read "Of the total 1343 women 300 (22.34%) were diagnosed with GDM.......gestation"

Introduction -

Do you really mean "any degree of glucose intolerance" There are specific criteria for GDM.

Suggest With "changes" in lifestyle and nutrition as some of the changes such as reduced physical activity are not "improvements"

In the last paragraph suggest you refer to reference 26 and also say in what way what you are investigating is new. Its all in the discussion but can be concisely noted here too.

Methods

Clinical and laboratory measurements - paragraph 1. Suggest you omit the last sentence "The BMI>24kg/m2 subgroup... in the results." as its confusing here. Its all clear in the results section.

Statistical comparisons in tables. In Supplementary table 1 the three sets of comparison are clear. In Supplementary table 2 its less clear as there is only one set of p values. Please make this clearer in the methods section and or the text when you refer to these tables. 

Results Lines 176-179 Suggest you leave out "which indicated that...during pregnancy." Alternatively you can reword the whole sentence saying that weight gain was less in the obese women.

The discussion is good but you need to make it very clear how this can be used in practice - are insulin assays readily available ?

In the conclusion suggest you change "to avoid GDM" to "to reduce GDM"

References Have a look at the recent paper by Shuoning Song et al in J Gynaecol Obstet in July 2022 on HOMA IR and GDM and include it if you think that helps,

Author Response

Thanks for your precious comments. We have amended the manuscript and retained track-changes. Here all the comments:

  1. The term “pre-pregnancy BMI”in this manuscript has been changed to “first trimester BMI (FT-BMI)”.
  2. In the abstract, resultshave been rephrased, and the revised sentence is as follows: “Of the total 1343 women, 300 (22.34%) were diagnosed with GDM in the 24-28 weeks of gestation.”
  3. In the introduction, the description of GDM has been changed. “Gestational diabetes mellitus (GDM) is a conditionof glucose intolerance that is first recognized during pregnancy, which is diagnosed during the second trimester.”
  4. In the introduction, the term “improvements” has been changed to “changes”.
  5. In the last paragraph of introduction, we added somecontent and illustrate the importance of our study. We emphasized that there is not a threshold of HOMA-IR based on different BMI for GDM diagnosis, and the cutoff value of HOMA-IR for Chinese women has not been unified.
  6. In the method--clinical and laboratory measurements, the sentence--“The BMI≥24kg/m2subgroup included the BMI≥28kg/m2 subgroup in the results.” has been deleted.
  7. In Supplemental Table 2, we marked at the bottom of the table that “TheP value shown in the table was among the four subgroups.” Because the indices of glucose and lipid profiles were significantly different among the four subgroups (P<0.001), we did not further distinguish the differences in pairs.
  8. In the second paragraph of the results, we rewrote the lase sentence. “The gestational weight gain (GWG) was the highest (13.0 kg [11.0–16.0]) in normal weight subgroup but thelowest (8.5 kg [4.6–12.0]) in obesity subgroup, which indicated that weight gain was greater in the normal weight women and less in the obese women.” 
  9. In the discussion, we emphasized that “The standardized insulin assay was not available in the past few years, however, the technique of serum insulin test has become more and more popular for clinical use in recent years, and the diagnosis of GDM is easier and earlier by HOMA-IR in early pregnancy compared with75-g OGTT during the second trimester.”
  10. In the discussion, wehave changed “to avoid GDM” to “to reduce GDM”.
  11. In the discussion, we included the recent paper by Shuoning Song et al in J Gynaecol Obstet in July 2022 on HOMA IR and GDM. We added it in the third paragraph in the discussion. “Song et al. provided a novel surrogate index of HOMA-IR, however, the formula includedpre-pregnancy BMI, FBG and lipid profiles, which was relatively complicated and with poor practicability.

Reviewer 2 Report

The study is well-written and in my opinion is pertinent to elucidate the contribution of weight in gestational diabetes mellitus (GDM).  As authors say, HOMA-IR may be affected by race. Is pertinent and necessary investigate relationship between HOMA-IR and body weight in pregnant women, according their initial weight.

This is a prospective observational study about a topic of interest, the association of BMI and gestational diabetes.

A clear opportunity to carry out the study is to know other races apart from the Caucasian, as the authors point out.

My comments are for a better explanation to readers of some methodology aspects: 

On the other hand, perhaps it would be opportune to mention that even today there are differences in the diagnosis of GDM, to clarify the specific criteria used in the study (for example, in Spain, and in other countries with high rates of central obesity, screening is routinely used with 75 g of glucose (O'Sullivan test) but confirmation is with an OGTT of 100 g of glucose. This is probably associated to the mean BMI of the race.

Is of great interest the first row of Supplemental table 1, where is depicted the incidence of GDM in every class of BMI. The incidence is very high in women whit more than 28 of BMI, highlitghing the importance of weight in GDM.  

Another thing I would like to ask: has the data collection been influenced at some point by the COVID-19 pandemic? Since it is plausible that pregnant women with COVID could have changes in the gestational period, in the incidence of GDM or both (several bibliographic references advocate this). If so, it should be mentioned.

Author Response

Thanks for your precious comments. We have amended the manuscript and retained track-changes. Here all the comments:

  1. Because the criteria of GDM has not been unified worldwide, we added in the last paragraph in the introduction that “So far, there are still differences in the diagnosis of GDM. One-step strategy of 75-g oral glucose tolerance test (OGTT) is adopt in ourcountry, while in other countries (especially in European countries) the diagnosis of GDM is confirmed by the two-step strategies of 100-g OGTT.”
  2. The incidence of GDM is high in obese women (BMI≥28kg/m2), and we emphasized in the discussion that “HOMA-IR is high correlated with body weight, so that the cutoff value of HOMA-IR for identifying GDM should be distinguished according to different FT-BMI values.
  3. The data collection in our study was influenced by the COVID-19 pandemic, several participants failed to be followed up to delivery. However, allthe participants in our study were not infected by COVID-19, so the incidence of GDM was not affected. We explained this in the limitations. “All participants in our study were not infected with COVID-19 under the dynamic zero-COVID policy and strategies executed by our country, and all participants were followed up to 24-28 weeks of gestation, which did not affect the results.”

Reviewer 3 Report

The manuscript is well constructed. 

General comment.  Scientifically speaking, the data are interesting but not innovative or practice-altering: it is highly unlikely that two fasting laboratory parameters (insulin and glucose, plus a calculation) will be widely used as a clinical predictor, particularly if you see the modest ROC predictions (figure 1).

My particular comments are:

- line 83: without impaired glucose tolerance or diabetes.  In most cases, this has not been properly tested before pregnancy.  I would suggest: without KNOWN impaired ...

- line 102: the fasting blood glucose...   How did the authors ensure that all patients were properly fasting at the time of the booking visit?  In many settings, this provision would be next to insurmountable.  

Author Response

Thanks for your precious comments. We have amended the manuscript and retained track-changes. Here all the comments:

  1. We have changed the sentencein the method--study participants. “without known impaired glucose tolerance or diabetes before pregnancy.”
  2. We asked the participants inour study to fast for 10- to 12- hours before the collection of blood samples, and we explained this in the method--clinical and laboratory measurements: “All fasting blood samples in our study were collected in the morning after a 10- to 12-h overnight fast followed by the doctor’s advice.”